# Two-Faced Roles of Tumor-Associated Neutrophils in Cancer Development and Progression

**DOI:** 10.3390/ijms21103457

**Published:** 2020-05-14

**Authors:** Naofumi Mukaida, So-ichiro Sasaki, Tomohisa Baba

**Affiliations:** Division of Molecular Bioregulation, Cancer Research Institute, Kanazawa University, Kakuma-machi, Kanazawa, Ishikawa 920-1192, Japan; s_sasaki@staff.kanazawa-u.ac.jp (S.-i.S.); sergenti@staff.kanazawa-u.ac.jp (T.B.)

**Keywords:** chemokine, CXCR2, CXCR4, granulocyte-colony stimulating factor (G–CSF), interleukin-17, myeloid-derived suppressor cell

## Abstract

Neutrophils are the most abundant circulating leukocytes in humans. Neutrophil infiltration into tumor tissues has long been observed but its roles have been ignored due to the presumed short life cycle and metabolic incompetence of neutrophils. Recent advances in neutrophil biology research have revealed that neutrophils have a longer life cycle with a potential to express various bioactive molecules. Clinical studies have simultaneously unraveled an increase in the neutrophil–lymphocyte ratio (NLR), a ratio of absolute neutrophil to absolute lymphocyte numbers in cancer patient peripheral blood and an association of higher NLR with more advanced or aggressive disease. As a consequence, tumor-associated neutrophils (TANs) have emerged as important players in tumor microenvironment. The elucidation of the roles of TANs, however, has been hampered by their multitude of plasticity in terms of phenotypes and functionality. Difficulties are further enhanced by the presence of a related cell population—polymorphonuclear leukocyte (PMN)-myeloid-derived suppressor cells (MDSCs)—and various dissimilar aspects of neutrophil biology between humans and mice. Here, we discuss TAN biology in various tumorigenesis processes, and particularly focus on the context-dependent functional heterogeneity of TANs.

## 1. Introduction

Neutrophils, the first-line sentinel to tissue damage, are the most abundant circulating leukocytes, representing 50% to 70% of all circulating leukocytes in humans [1]. The production of neutrophils is estimated to be 10^11^ and 10^7^ cells per day at steady state in humans and mice, respectively. Tumor tissues are usually infiltrated with various types of leukocytes including neutrophils, macrophages and lymphocytes [2]. Compared with macrophages and lymphocytes, less attention has been paid to tumor-associated neutrophils (TANs) until recently, because neutrophils were previously presumed to be short-lived and contain minimal amount of mRNA to express bioactive molecules [3].

Evidence is, however, accumulating to indicate that in response to various external cues, neutrophils can express abundantly various cytokines [4,5] and can release various types of bioactive molecules stored in their granules [6]. Moreover, a recent study estimated the half-life of human neutrophils up to 5.4 days based on in vivo labeling human neutrophils with the use of ^2^H_2_O [7], in contrast to previously estimated half-life of less than one day based on ex vivo labeling or manipulation [8]. Thus, neutrophils can persist for a period of time sufficient to allow them to express and release bioactive molecules in tumor microenvironment. Moreover, the important roles of neutrophils in tumorigenesis has been suggested by the observations that patients with various types of cancers exhibited an increase in the neutrophil–lymphocyte ratio (NLR), a ratio of absolute neutrophil to absolute lymphocyte numbers in peripheral blood and that NLR was elevated in high-risk patient population with more advanced or aggressive disease [9]. Additionally, a large-scale meta-analysis of expression signatures revealed that TAN signatures were associated with poor disease outcomes in diverse types of solid tumors including breast and lung adenocarcinomas [10]. With these observations, TANs have emerged as an important player in tumor microenvironment.

Accumulating evidence indicates the association of a high density of TANs with poor prognosis of cancer patients [11], but this association did not apply to several types of cancers, such as squamous cell lung cancer [12] and colorectal cancer [13]. These context-dependent distinct roles of TANs may mirror their diverse actions under various tumor microenvironments. In this review article, after briefly summarizing the biology of neutrophils, we will discuss the functional heterogeneity of TANs in various tumorigenesis processes and the possible anticancer strategies targeting the diversity of TANs.

## 2. Differentiation and Plasticity of Neutrophils

Human neutrophils transit from bone marrow into circulation within six to seven days after the last cell division [14]. On the basis of morphologic features including cell size, nuclear condensation and granule content, their different stages were classified from granulocyte–monocyte progenitors (GMPs), through granulocyte-committed precursors comprising myeloblasts, promeylocytes and myelocytes, to a postmitotic or transition pool of metamyelocytes, band cells and segmented neutrophils [15]. Recent analysis on the transcriptome at the single-cell level revealed that hematopoietic stem cells (HSCs) differentiate into each lineage including granulocyte one, in a continuous way [16]. A cell cycle-based and mass cytometric analyses revealed three neutrophil subsets in mouse bone marrow—a proliferative neutrophil precursor—which can differentiate into non-proliferating immature neutrophils and mature neutrophils [17]. These three subsets can be identified in mouse and human, based on the distinct expression patterns of surface markers (Table 1) [18]. Under the influence of several colony-stimulating factors (CSFs), particularly granulocyte (G)-CSF, neutrophil precursors are generated from GMPs depending on the action of a transcription factor, C/EBPα, while their subsequent differentiation towards non-proliferative neutrophils is dependent on C/EBPε and PU.1 [19,20].

Mature neutrophils are released from bone marrow to circulation mainly by the action of G–CSF, interleukin (IL)-17 and neutrophilic CXC chemokines with ELR motif, such as CXCL8, CXCL1, CXCL2, CXCL3 and CXCL5 [21,22]. As a major neutrophil chemoattractant, humans utilize CXCL8, which can bind with CXCR1 and CXCR2 [23]. On the contrary, mice and rats lack *CXCL8* and *CXCR1* genes and as a consequence, their neutrophils utilize CXCR2 which bind other ELR^+^ CXC chemokines, CXCL1, CXCL2, CXCL3 and CXCL5 [23]. Nevertheless, a proportion of neutrophils are localized in marginated pool in lung, liver and spleen and can be mobilized back into circulation under stressed conditions, particularly infection [24,25]. In response to tissue injury, neutrophils extravasate from circulation into the tissues under the coordinate regulation of various adhesion molecules and chemokines, particularly CXC ones [26]. As tissue injury is resolved, extravasated neutrophils mostly die in the tissue and are cleared by macrophages, but some of them reenter the vasculature, the phenomenon called as reverse neutrophil migration [27]. Moreover, reverse-transmigrating neutrophils are phenotypically and functionally distinct from non-reverse transmigrating ones [28], indicating the heterogeneity among the neutrophils which extravasate inflammatory tissues. Aging neutrophils exhibit enhanced expression of a chemokine receptor, CXCR4 and migrate back to bone marrow, where they are retained by the action of stroma cell-derived CXCL12, a ligand for CXCR4 and are eventually cleared by macrophages [24,29]. A recent study unraveled that neutrophils actively infiltrate most organs under physiological conditions except brain and gonad and that they can have distinct roles in each organ: inhibition of cytokine production by intestinal macrophages, regulation of diurnal transcription in lungs and control of the metastatic invasion in lungs [30]. Moreover, there exist two distinct phenotypically and functionally distinct neutrophil subsets in spleen [31]. These observations would indicate the heterogeneity among neutrophils even under physiological conditions.

Neutrophil migration into tumor tissues is also regulated by the combined action of G–CSF, IL-17 and neutrophilic ELR^+^ chemokines [21,22] (Figure 1). Once TANs accumulate in tumor tissues, they exhibit functional heterogeneity and the presence of two polarized states, N1 and N2, were proposed [32], analogous to helper T lymphocyte and monocyte polarization [33,34]. N2-like TANs can exhibit pro-tumorigenic activities in the same way as M2 macrophages and transforming growth factor (TGF)-β blockade changed their phenotype to N1 exhibiting an enhanced cytotoxicity to tumor cells, similarly as M1 macrophages do [32]. Similarly, type I interferon (IFN) induced the phenotype change from N2- to N1-like one in both mouse and human neutrophils [35]. In an autochthonous mouse uterine cancer model, infiltrated neutrophils exhibited N2-like phenotypes and released neutrophil elastase (NE) to promote tumor growth, and improved tumor oxygenation converted neutrophil phenotypes to N1-like one with a higher ability to kill tumor cells by producing NADPH oxidase-derived reactive oxygen species (ROS) and matrix metalloproteinase (MMP)-9 [36]. These observations would imply that TANs can change their phenotypes and functions in response to the cues from tumor microenvironment.

Myeloid-derived suppressor cells (MDSCs) exhibit immunosuppressive functions and express both granulocyte and macrophage markers [37]. They can be further subdivided into three subsets, early, monocytic- and polymorphonuclear (PMN)-MDSCs, based on their expression of cell surface markers (Table 2). Mouse PMN–MDSCs are defined by the Ly6G marker, but the proposed candidate markers in humans, CD15 and CD66b, are expressed in human immature and mature neutrophils [38], thereby precluding the discrimination of human PMN–MDSC from neutrophils solely based on their phenotypes. Human PMN–MDSCs have a distinct gene expression profile, characterized by the enhanced expression of genes associated with endoplasmic reticulum (ER) stress, particularly *lectin-type oxidized low-density lipoprotein receptor 1* (*LOX-1*) gene, compared with conventional neutrophils [39]. Moreover, induction of ER stress in neutrophils augmented *LOX-1* gene expression and converted these cells to immunosuppressive PMN–MDSCs. Thus, PMN–MDSCs may be another face of TANs.

Several groups reported the presence of an additional neutrophil subpopulation, low-density neutrophils (LDNs) in cancer patients or tumor-bearing mice, and their immunosuppressive activities [40,41]. However, it has been reported that neutrophil density can be modulated by in vitro stimulation at sample preparation [42].

## 3. TANs in Tumor Development and Growth 

Neutrophils can release various molecules from their granules and cytoplasm and can produce various cytokines and chemokines under stressed conditions [43]. These molecules can affect the tumor microenvironment by acting on both tumor cells and stroma cells (Figure 2).

Neutrophils store a large amount of myeloperoxidase (MPO) in their azurophilic granules [44]. MPO can abundantly generate HOCl, thereby enhancing the formation of 3-(2-deoxy-β-d-*erythro*-pentofuranosyl)pyrimido [1,2-α] purin-10(3*H*)-one (M_1_dG) [45]. Moreover, intratracheal injection of lipopolysaccharide (LPS) induced intrapulmonary neutrophil infiltration together with augmented M_1_dG formation, and neutrophil depletion decreased intrapulmonary MPO activities and M_1_dG formation. Thus, neutrophil-derived MPO catalyzed HOCl formation, thereby inducing mutagenesis, the first step of carcinogenesis, in adjacent epithelial cells (Figure 2).

Cancer stem cells (CSCs) represent a minor subset of malignant cells, with unlimited self-renewal and differentiation ability and are responsible for tumor aggressiveness, tumor heterogeneity, metastasis and resistance to antitumor treatments [46]. TANs secreted a large amount of bone morphogenic protein (BMP)-2 and TGF-β2 to elicit microRNA (miR)-301-3p expression in human hepatocellular carcinoma (HCC) cells [47]. miR-301-3p suppressed the expression of cylindromatosis lysine 63 deubiquitinase (CYLD) and limbic system–associated membrane protein, and conferred cancer stem-like phenotypes on HCC cells. Moreover, since CYLD can act as a negative regulator of NF-κB pathways [48], depressed CYLD expression resulted in aberrant activation of NF-κB, thereby inducing HCC cells to express abundantly a neutrophilic chemokine, CXCL5, which can attract a large number of neutrophils [47]. Thus, a positive feedback loop may exist between TANs and cancer stem-like cells in HCC.

Neutrophils release proteinases with potent tumorigenic activities, NE and MMPs. Once being released, NE could promote lung carcinogenesis [49], by activating phosphatidylinositol 3-kinase (PI3 K) to augment cancer cell growth through degrading insulin receptor substrate-1 in lung cancer cells and increasing the interaction between PI3 K and platelet-derived growth factor (PDGF) receptor (R) [50].

Evidence is accumulating to indicate the crucial involvement of MMPs in tumor angiogenesis because they can promote the degradation of the vascular basement membrane, and can remodel the extracellular matrix (ECM), and induce subsequent endothelial cell migration and proliferation [51]. Indeed, neutrophil-derived MMP-9 could enhance tumor angiogenesis [52,53] and cancer cell intravasation [53], thereby facilitating tumor progression.

CXCL8 and its related ELR^+^ CXC chemokines are a major regulator of neutrophil migration but they have also diverse impacts on multiple aspects of cancer cells, including their proliferation, survival and migration [54]. ELR^+^ CXC chemokines are produced by various types of cells present in tumor tissues, such as cancer cells, endothelial cells, fibroblasts and macrophages [54]. Moreover, human neutrophils can in vitro produce CXCL8 upon co-culture with cancer cell lines [55,56]. Thus, TANs can produce CXCL8 and other ELR^+^ CXC chemokines, thereby further augmenting neutrophil migration, angiogenesis and tumor growth (Figure 2). This notion may be supported by the observation that CXCR2 inhibition suppressed tumor development and progression together with decreased TANs in several mouse tumor models [57,58].

SMAD4, a downstream mediator of the TGF-β signaling superfamily, can act as a tumor suppressor in colon carcinogenesis while its loss is observed in 30% to 40% of colorectal cancer cases and is associated with a poor prognosis [59,60]. Consistently, mice which lack *Smad4* gene selectively in colon epithelium, exhibited accelerated colon carcinogenesis together with increased neutrophil infiltration [61]. Neutrophil infiltration was induced by either CCL15 [62] or CXCL1 and CXCL8 [63], the chemokines that were produced by Smad4-negative colon cancer cells. The attracted neutrophils expressed CXCL1 and CXCL8, MMP-2 and MMP-9, thereby inducing tumor angiogenesis and subsequent tumor progression and metastasis [63].

Neutrophils can produce various angiogenic or growth factors [4,5,43] (Figure 2). Vascular endothelial growth factor (VEGF) has a central role in tumor angiogenesis [64] and is stored intracellularly in human neutrophils [65]. Moreover, both human and mouse neutrophils can in vitro release VEGF in response to various stimuli such as tumor necrosis factor (TNF)-α, GM–CSF, G–CSF and neutrophilic chemokines [65,66,67,68]. The absence of IFN-β augmented intratumoral infiltration of CXCR2-expressing neutrophils [69], which abundantly produced VEGF and MMP-9 [70] and as a consequence, neutrophil depletion reduced tumor angiogenesis [70]. The expression of another angiogenic factor, angiopoietin 1 (Ang1), in human neutrophils has been documented [71]. However, it remains still elusive on the roles of neutrophil-derived VEGF and Ang1 in tumor angiogenesis and subsequent tumor growth. In collaboration with VEGF, fibroblast growth factor 2 (FGF)-2 can promote tumor angiogenesis [72]. In hepatic metastatic sites, TANs expressed abundantly FGF-2, which accelerated angiogenesis and eventually promoted liver metastasis growth [73].

Bv8/prokinectin 2 (Prok2) and its related molecule, Prok1, are characterized with a five-disulfide bridge and govern gastrointestinal integrity and neuronal development by acting on their specific G-protein-coupled receptor, Prokr1 and Prokr2 [74]. In human lung adenocarcinoma tissues, Bv8/Prok2 was detected in infiltrating neutrophils [75]. Tumor cell implantation induced G–CSF expression in mice and eventually augmented the expression of Bv8/Prok2 in CD11b^+^Gr1^+^ neutrophils [76]. Moreover, anti-Bv8/Prok2 antibodies reduced neutrophils in peripheral blood and tumor sites and suppressed tumor angiogenesis, thereby inhibiting tumor growth. The same group further claimed that enhanced Bv8/Prok2 expression in neutrophils could account for frequently observed refractoriness to anti-VEGF therapy in tumor-bearing mice [77].

Oncostatin M (OSM) is a member of IL-6 family cytokines with diverse activities [78], and it can stimulate angiogenesis by enhancing FGF-2 expression [79] and can in vitro induce mesenchymal- and cancer stem cell-like phenotypes in cancer cells [80]. Human breast cancer cell lines produced GM–CSF, which could induce human neutrophils to secrete OSM [81]. The produced OSM boosted VEGF expression in breast cancer cell lines and increased their detachment and invasive capacity, suggesting that tumor progression can be mediated by neutrophil-derived OSM (Figure 2). This assumption may be substantiated by the observations on skin squamous cancer (SCC), where OSM was detected mainly in TANs and OSM depletion partially reduced the sizes of mouse SCC [82].

TGF-β was also detected in TANs, which were located at the invasive edges of human pancreatic ductal adenocarcinoma (PDAC) tissues and was presumed to contribute to the development of fibrotic changes in PDAC [83]. Moreover, TAN-derived TGF-β could induce epithelial–mesenchymal transition (EMT), the first step of cancer cell invasion, in human lung cancer tissues [84] (Figure 2).

Hepatocyte growth factor (HGF) is produced mainly by fibroblasts in tumor microenvironment, and mediates various tumorigenic processes including cell proliferation, survival, invasion and angiogenesis, by acting on its specific receptor, Met [85]. In human bronchioloalveolar-subtype lung adenocarcinoma tissues, HGF and Met were detected in TANs and in cancer cells, respectively [86], suggesting that tumor progression can be provoked by TAN-derived HGF. This assumption may be supported by the observations that HCC cell-derived GM–CSF-activated TANs to produce HGF and that the treatment with anti-neutrophil or anti-HGF antibodies reduced tumor formation [87]. On the contrary, TNF-α induced Met expression in neutrophils, which could release nitric oxide upon HGF stimulation, thereby killing tumor cells [88]. These observations suggest that TANs can exhibit context-dependent distinct roles in carcinogenesis through the actions of the HGF–Met axis.

TANs can have similar Janus-faced roles in colon cancer model induced by the combined treatment of azoxymethane (AOM) and dextran sulfate sodium (DSS). In this model, a massive neutrophil infiltration was observed in colon cancer tissues [89] and anti-neutrophil antibody treatment decreased TANs which promoted carcinogenesis by providing MMP-9 [90]. On the contrary, genetically neutropenic mice were more susceptible to AOM/DSS-induced colon carcinogenesis in association with increased bacteria numbers in tumors [91], indicating that neutrophils can restrict bacteria growth and subsequently delay tumor development. This counteracting action of neutrophils may account for improved survival of stage II colorectal cancer patients with high levels of TANs [13].

Similar two-faced roles were observed on neutrophil extracellular traps (NETs) which are release by neutrophils and are composed of DNA coated with histone, NE, MPO, MMP-9 and cathepsin G [92]. NETs can inhibit melanoma cell migration and their proliferation, probably with the use of MPO [93]. On the contrary, tumor cells can induce the release of NETs [94], which can further promote tumor progression by inducing blood clot formation [95]. Moreover, NETs can further provide NE and/or MMP-9 to promote tumor cell growth [96]. NETs can further trap circulating tumor cells (CTCs) in vitro and in vivo [97,98], thereby facilitating their migration into distant organs.

## 4. TANs in Tumor Immunity 

Neutrophils were presumed to be incapable of directly killing cancer cells without any additional stimuli [99]. However, TNF-α treatment induced neutrophils to kill tumor cells by generating ROS [100]. Similarly, radiation therapy to syngeneic mouse tumor models induced a massive intratumoral infiltration of neutrophils, which induced tumor apoptosis with the use of ROS [101]. Moreover, neutrophil-derived superoxide induced calcium influx in tumor cells by acting on transient receptor potential cation channel, subfamily M, member 2 (TRPM2), expressed in cancer cells and eventually cell lysis [102]. Additionally, IL-2 augmented neutrophil cytotoxicity against tumor cells by inducing nitric oxide generation [103]. Moreover, IFN-γ and IL-2 could induce neutrophils to express granzyme B, which was directly cytotoxic to tumor cells [104]. Neutrophils can additionally exert antibody-dependent cell cytotoxicity (ADCC) against antibody-coated tumor cells [105] through trogoptosis, which can be further augmented by blocking the CD47-SIRPα interactions [106]. Superficial bladder cancer is frequently treated with *Mycobacterium bovis* bacillus Calmette–Guérin (BCG), which can induce neutrophils to express tumor necrosis factor-related apoptosis-inducing ligand (TRAIL) with a direct tumor killing activity [107] (Figure 3).

Neutrophils can have positive effects on adaptive immunity including tumor immunity by directly interacting with lymphocytes and dendritic cells (DCs) or by producing bioactive molecules that affect them [108] (Figure 3). Neutrophil depletion dampened both the priming and effector phases of immunity against tumor antigens [109]. Neutrophils can in vivo take up and present exogenous antigen into major histocompatibility antigen (MHC) class I molecules to prime naïve T cells [110] and can present antigens also to memory CD4^+^ lymphocytes in MHC class II-dependent manner [111]. Moreover, GM–CSF can in vitro induce neutrophils to differentiate into a hybrid population showing dual properties of both neutrophils and DCs, with a potent antigen-presenting ability [112]. Similar hybrid population was also detected in early stage human lung cancer [113]. Given the capacity of neutrophils to migrate into draining lymph nodes using a chemokine receptor, CCR7 [114], they may contribute to the establishment of tumor immunity by using their antigen presenting ability. This assumption was supported by the study on the draining lymph nodes obtained from patients with early stage lung cancer [115]. This study unraveled that TANs could stimulate T cell proliferation and IFN-γ release and that the cross-talk between TANs and activated T cells led to enhanced costimulatory molecule expression on the neutrophil surface, thereby boosting T cell proliferation in a positive-feedback manner.

Similar crosstalk between neutrophils and immune cells was observed on other neutrophil-mediated antitumor effects (Figure 3). *G–CSF* gene transduction into murine adenocarcinoma cells, elicited neutrophil-dependent tumor regression in vivo [116] and this regression was associated with IFN-γ, which was produced mainly by CD8^+^ lymphocytes through the interaction with neutrophils [117]. IL-17-expressing CD4^+^ T cells were present in human esophageal squamous cell carcinoma (ESCC) tissues [118] and IL-17 could attract neutrophils by provoking the production of neutrophilic chemokines, CXCL2 and CXCL3, in ESCC cells and simultaneously activate neutrophils to express cytotoxic molecules, including MPO, TRAIL and IFN-γ, thereby inducing cancer cell death [119].

Tumor cell-derived ELR^+^ neutrophilic chemokines induced neutrophils and PMN–MDSCs to form NETs, which can wrap and coat tumor cells, thereby protecting tumor cells from CD8^+^ T cell- and natural killer (NK) cell-mediated cytotoxicity [120]. Additionally, neutrophils can directly suppress T cell-mediated immunity. IL-1β elicited IL-17 expression from γδ T cells, which promoted TAN expansion in mammary cancer-bearing mice, in concert with G–CSF, and expanding TANs suppressed cytotoxic CD8^+^ T cell functions [121]. The interaction between immune checkpoint molecules, PD-1 and PD–L1, may be responsible for neutrophil-mediated immune suppression. In human hepatocellular carcinoma (HCC) tissues and HCC-bearing mice, PD–L1 and PD-1 were expressed in peritumoral neutrophils and T lymphocytes, respectively, and PD–L1^+^ neutrophils from patients with HCC effectively suppressed the proliferation and activation of T cells by utilizing the PD-1/PD–L1 interactions [122].

Several enzymes are presumed to exert immune suppressive activities (Figure 3). Arginase 1 (Arg1) can also impair T cell functions in tumor-bearing mice by depleting L-arginine [123]. Non-small cell lung cancer (NSCLC) cell lines produced abundantly CXCL8, which can induce TANs to release Arg1, thereby inducing extracellular L-arginine depletion and subsequent immune suppression [124]. Moreover, ROS generated by neutrophil-derived MPO, can inhibit NK cell activity against tumor cells [125]. Furthermore, A serine protease, proteinase 3 (P3), is localized in azurophil granules and on the cell surface membrane of neutrophils, and membranous P3 could inhibit T cell proliferation through its direct interaction with low density lipoprotein receptor-related protein 1 [126].

TANs can suppress tumor immunity by recruiting immune suppressive cells (Figure 3). In syngeneic mouse tumor models, TANs in tumor-bearing mice secreted CCL17 to induce regulatory T cell (Treg) infiltration into tumor sites, thereby boosting tumor growth [127]. Consistently, in human hepatocellular carcinoma tissues, TANs expressed abundantly CCL17 and CCL2, which could attract Tregs and macrophages, respectively, and depletion of neutrophils enhanced the sensitivity to sorafenib in mouse HCC models, as evidenced by reduced tumor growth [128]. In acute graft-versus-host disease (GVHD), G–CSF treatment generated a population of activated neutrophils, which can dampen GVHD by producing IL-10 and attracting Tregs [129].

Diverse immune activities of TANs can be ascribed to the presence of functional dichotomy of TANs between N1- and N2-like ones, which can exhibit antitumorigenic and pro-tumorigenic functions, respectively [32]. N2-like phenotype can be maintained under the influence of tumor microenvironment, particularly hypoxia [36], which is more prevalent in advanced tumor sites than early ones [130]. Thus, TANs may exhibit distinct immune functions triggered by tumor environmental cues.

A TAN-related cell population, PMN–MDSCs were increased in tumor sites and peripheral blood of several types of cancers including human NSCLC [131], PDAC [132] and glioma [133] and can in vitro suppress T cell functions with the use of Arg-1. Moreover, parenchymal PMN–MDSC in human renal cell cancer, have a positive correlation with IL1-β, CXCL8, CXCL5 and CCL3 and CXCR2 blockade reduced tumor weight with enhanced CD4^+^ and CD8^+^ T-cell infiltration in renal cell carcinoma-bearing mice [134]. Oncogenic activation of KRAS in colorectal cancer cells, depressed the expression of IRF2 with a capacity to depress CXCL3 expression and eventually enhanced expression of CXCL3 which promoted intratumoral migration of PMN–MDSCs expressing CXCR2, a receptor for CXCL3 [135]. Moreover, the resultant PMN–MDSC infiltration was associated with the refractoriness to immune checkpoint therapy.

## 5. TANs in Tumor Metastasis

Metastasis proceeds through multiple processes [136]. Cancer cells at the primary site invade locally the adjacent tissues and intravasate into systemic circulation including blood and lymphatics, after undergoing EMT (Figure 4). CTCs extravasate through endothelial walls into the parenchyma of distant organs and form micrometastatic colonies, which subsequently develop into clinically detectable metastatic lesions [137]. The important roles of neutrophils in metastasis have been substantiated by the reduction of metastasis by neutrophil depletion in several animal models [138].

Hu and colleagues revealed negative association of intratumoral CD66b^+^ neutrophil numbers with tumoral E-cadherin expression in human lung adenocarcinoma tissues [84]. They further demonstrated that co-culture with neutrophils promoted EMT of cancer cells and enhanced their migration in vitro, but without any identification of a molecule(s) responsible for the process. Subsequent study demonstrated that neutrophil-derived tissue inhibitor of metalloproteinase (TIMP)-1 could induce EMT in human breast cancer cells and could eventually promote metastasis [139]. Li and colleagues demonstrated that TAN numbers correlated with tumor stage and could predict poor prognosis in human gastric cancer patients [140]. Moreover, TANs were localized mainly at the invasive margin of human gastric cancer tissues and expressed abundantly IL-17, which enhanced the migration, invasion and EMT of gastric cancer cells, through the activation of JAK2/STAT3 pathways [140].

CTCs are highly vulnerable to anoikis, due to loss of adhesion to extracellular matrix and hemodynamic stress [141]. CTCs associated with neutrophils, were occasionally isolated from human patients with breast cancer and mouse models [142]. Moreover, CTCs associated with neutrophils, exhibited highly metastatic efficiency with robust expression of proliferation-related genes, compared with CTCs lacking the association with neutrophils. Thus, neutrophils can provide CTCs with the signals to facilitate cell cycle progression and evade anoikis, thereby promoting metastasis (Figure 4).

Tumor-derived factors and extracellular vesicles can transform the microenvironment (=soil) of distant organs to so-called pre-metastatic niche, which can support the outgrowth of incoming cancer cells (=seeds) [143]. In mice, systemic elevation of TIMP-1 levels increased hepatic CXCL12 levels, which promoted the intrahepatic recruitment of neutrophils [144]. Similarly, by acting TLR3, tumor exosomal RNAs induced alveolar epithelial cells to express abundantly neutrophilic chemokines, which attracted neutrophils into lungs [145]. Although precise mechanisms were not clarified, infiltrated neutrophils stimulated pre-metastatic niche formation and eventually augmented metastasis in both instances [144,145]. Mice with type I IFN receptor deficiency exhibited elevated serum G–CSF levels and enhanced CXCR2 expression on neutrophils [146]. These phenotypic changes boosted intrapulmonary infiltration of neutrophils, which expressed pro-metastatic molecules such as Bv8/Prok2, MMP-9, S100A8 and S100A9, and eventually augmented metastasis formation in murine experimental lung metastasis models. Lack of IFN-β gene further delayed apoptosis of TANs due to augmented PI3 kinase activation [147]. Additionally, before tumor cell arrival, neutrophils provided lungs of mice bearing mammary adenocarcinomas with MMP-9 to promote vascular remodeling, thereby forming pre-metastatic niche [148] (Figure 4).

Neutrophils can bind directly with CTCs via Mac-1-mediated interaction and can act as a bridge between cancer cells and liver parenchyma, thereby promoting the extravasation of cancer cells into liver parenchyma [149]. Moreover, sustained Notch1 activation in lung endothelial cells induced endothelial senescence which was associated with enhanced chemokine and vascular cell adhesion molecule-1 expression [150]. This eventually promoted adhesion of both tumor cells and neutrophils to the endothelium and eventually extravasation and lung colonization of tumor cells (Figure 4).

Neutrophils can promote cancer cell invasion and extravasation by utilizing NETs (Figure 4). NETs could trap CTCs in vitro under static and dynamic conditions and in vivo under infection [97]. Moreover, β1-integrin expressed on both cancer cells and NETs could mediate the adhesion of CTCs to NETs both in vitro and in vivo [98]. NET trapping was associated with enhanced formation of hepatic metastasis [97] through the interaction of colon cancer cells with NET-associated carcinoembryonic antigen cell adhesion molecule 1 [151]. A subsequent study reported that NET trapping was observed in clinical samples of triple-negative breast cancer patients as well as mouse model even in the absence of infection and that NET digestion with DNase I markedly reduced lung metastases in mice, indicating the crucial roles of NET in metastasis even in the absence of infection [152]. NET formation was also required for pre-metastatic niche formation in the omentum in mice bearing ovarian cancer cells and eventually facilitated implantation of ovarian cancer cells and their subsequent dissemination in peritoneal cavity [153].

G–CSF was produced by cancer cells as well as host resident cells in pre-metastatic lungs of mice bearing a mouse breast cancer cell line, 4T1 and induced neutrophil infiltration into lungs [154]. Infiltrated neutrophils provided cancer cells with Bv8/Prok2 with a potent mitogenic activity to promote cancer cell growth in lungs. An additional proteomic approach with a functional screen was conducted to elucidate the secretome of TANs in the same lung metastasis model using 4T1 and identified an iron-transporting protein, transferrin, as the TAN-derived major mitogen for tumor cells [155]. Moreover, cancer cell-derived GM–CSF induced transferrin expression in TANs through JAK1/2/STAT5β pathway activation. Neutrophil depletion or transferrin receptor deletion in cancer cells, reduced lung metastasis arising from orthotopic injection of 4T1 cells [155]. Likewise, we observed that 5-fluorouracil administration induced the expression of G–CSF and neutrophilic chemokines, CXCL1 and CXCL2, in cancer cells, by activating NF-κB pathways [156]. The produced cytokines induced the infiltration of neutrophils, which could facilitate lung metastasis formation by delivering Bv8/Prok2. Moreover, with the use of arachidonate 5-lipoxygenase, neutrophils at lung metastasis sites generated various leukotrienes (LTs) such as LTB_4_, LTC, LTD and LTE, to support tumor cell colonization by selectively expanding the subpopulation of tumor cells with a high tumorigenic potential [157].

However, the identity of neutrophil-derived molecule(s) crucially involved in metastasis and pre-metastatic niche formation still remains elusive [158]. Nevertheless, neutrophil migration is mostly governed by neutrophilic ELR^+^ chemokines such as CXCL1, CXCL2, CXCL3, CXCL5 and CXCL8, in concert with danger-associated molecules including DNA, high mobility group protein B1 (HMGB1), ATP and uric acid [159]. The expression of these chemokines are regulated at the transcriptional level by NF-κB [160], which is frequently activated in cancer cells [160,161]. Moreover, in response to oncogene activation or exposure to chemotherapeutic drugs, cancer cells are prone to undergo senescence, which can activate NF-κB pathway [162], thereby inducing the expression of neutrophilic chemokines. Thus, NF-κB activation may account for enhanced expression of neutrophilic chemokines and subsequent neutrophil-mediated metastasis. Alternatively, DNA demethylation can account for the enhanced expression of CXCL1 and CXCL8, thereby augmenting neutrophil-mediated metastasis, as proved in metastatic renal cancer cells [163].

A few studies reported anti-metastatic functions of neutrophils (Figure 4). The capacity of mature neutrophils to inhibit liver metastasis was documented without any information on the molecular mechanism [41]. In collaboration with G–CSF, tumor cell-derived CCL2 educated neutrophils to migrate into pre-metastatic lung and to produce HOCl, thereby inhibiting breast cancer seeding to lungs [164], probably by acting on TRPM2 expressed in tumor cells [102]. Moreover, TNF-α-induced HGF expression conferred nitric oxide-mediated anti-metastatic activity on neutrophils [88].

## 6. Future Perspective on Anticancer Treatment Targeting TANs

TANs can be a cellular target of anticancer treatment measures: neutrophil depletion, inhibition of neutrophil accumulation in tumor sites and modulation of neutrophil functional phenotypes. However, these measures have not been examined in clinical trials. For instance, the administration of anti-neutrophil antibody could deplete TANs in preclinical models [156], but it could simultaneously reduce circulating neutrophils, which have profound roles in diverse host responses to various insults, particularly bacterial infections [165]. Thus, neutrophil depletion can enhance susceptibility to bacterial infection and therefore, encounter difficulties in conducting a clinical trial.

As neutrophil migration is regulated by chemokine systems, particularly CXCR2 and CXCR4 and the G–CSF/IL-17 axis, targeting these molecules has been proposed as anticancer treatment strategy by reducing TAN accumulation at tumor sites. CXCR2 inhibition could reduce mouse lung tumorigenesis by reducing TAN accumulation [49] and could inhibit metastasis in mouse PDAC model [166]. In PDAC model, no direct proof on the effects of CXCR2 inhibition on TAN was provided, and therefore, CXCR2 blockade may act also on cancer cells and/or endothelial cells, which express CXCR2 [54]. CXCR4 inhibition was also examined for its anticancer effects. A CXCR4 inhibitor was effective to reduce tumor burdens and metastasis in small cell lung cancer-bearing mice although the effects on TANs were not described [167]. In colorectal cancer-bearing mice, resistance to anti-VEGF-R2 antibody treatment was associated with enhanced CXCR4 expression and its inhibition enhanced the effectiveness of anti-VEGF-R2 treatment, probably acting CXCR4-expressing inflammatory monocytes [168]. Thus, it still remains elusive on the effects of either CXCR2 or CXCR4 inhibition on TANs. On the contrary, several pre-clinical mouse studies indicate that G–CSF can support tumor progression and can promote tumor dissemination and metastasis, by mobilizing TANs, but human studies failed to unravel the association of enhanced G–CSF expression with increased tumor progression [169]. Moreover, G–CSF could enhance radiation-induced antitumor immune responses by activating TANs [101]. Similarly, IL-17 could induce the accumulation of TANs with either tumor promoting [22] or tumor killing activities [119] in a context-dependent manner. These observations would raise a question on the feasibility of blockade of the G–CSF/IL-17 axis as anticancer therapy strategy.

Given the plasticity of TANs as exemplified by pro-tumorigenic N2-like and antitumorigenic N1-like phenotype [32,35], the modification of TAN phenotype may be more desirable as an anticancer treatment. This assumption can be supported by a mathematical model, which predicted that antitumor efficacy could be increased when apoptotic cell death of N1 and N2 was depressed and augmented, respectively [170]. Instead of apoptosis induction, the phenotype conversion of TANs from N2- to N1-like ones can be achieved by oxygenation of tumor tissues [36].

Neutrophil functions at tumor sites have been elucidated mostly by using mice, but needless to say, there are several differences between mouse and human, in terms of tumor cell biology. Additionally, humans and mice do not share many aspects of neutrophil biology, from its migration steps to its effector functions [171]. Human, but not mouse L-selectin can bind E-selectin [172]. Human neutrophils utilize CXCL8 as a major neutrophilic chemokine and express two distinct receptors for ELR^+^ CXC chemokines, CXCR1 and CXCR2, whereas mice lack *CXCL8* and *CXCR1* gene [23]. Moreover, human neutrophils contain much higher levels of granular enzymes including MPO, lysozymes and alkaline phosphatase than mouse ones [173]. In contrast, mouse neutrophils can secrete IL-10 more abundantly than human ones, because chromatin structures around the promoter region of the *IL-10* gene differs between mouse and human [5]. Nevertheless, it is still unclear on most biologic aspects of TANs in human tumor tissues, particularly their specific functions in each type of human tumor tissue. As the conundrum will be solved with the use of recently developed analytical methods, such as single-cell RNAseq and/or mass cytometry, anticancer therapy targeting TANs will emerge as a promising strategy.

## Figures and Tables

**Figure 1 ijms-21-03457-f001:**
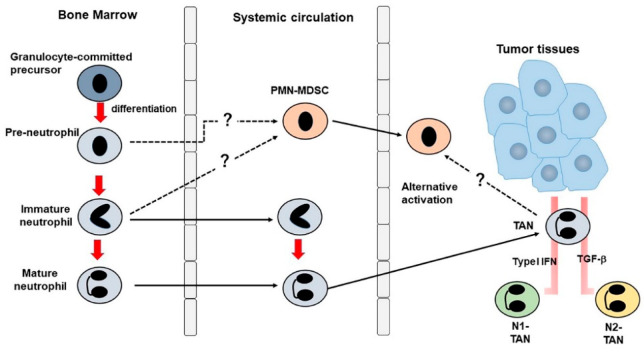
Presumed generation mechanism of tumor-associated neutrophils (TANs) and polymorphonuclear leukocyte (PMN)-myeloid-derived suppressor cells (MDSCs).

**Figure 2 ijms-21-03457-f002:**
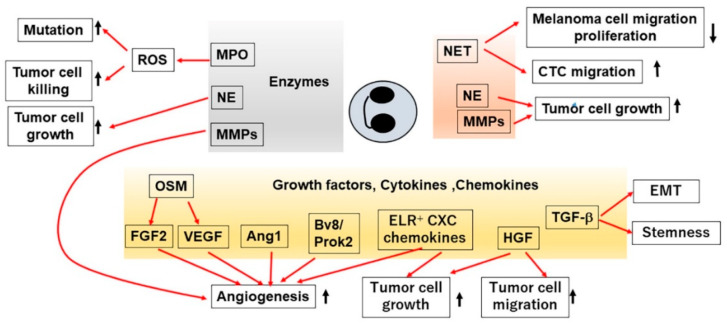
Presumed functions of TANs in tumor development and progression.

**Figure 3 ijms-21-03457-f003:**
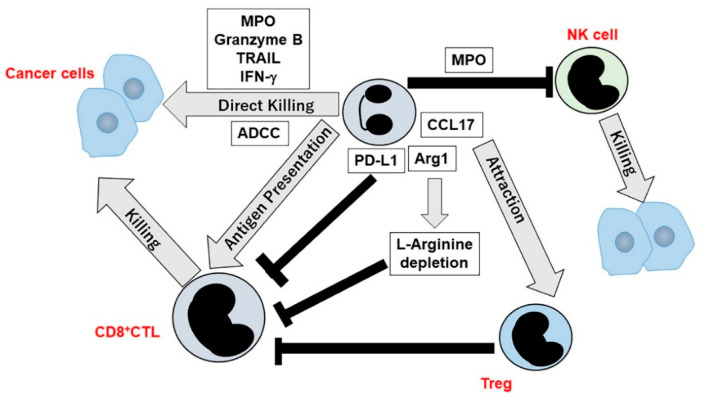
Effects of TANs on tumor immunity.

**Figure 4 ijms-21-03457-f004:**
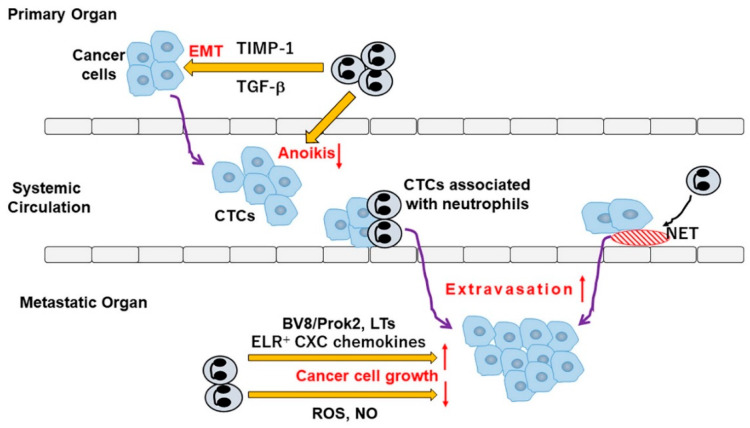
Presumed roles of TANs in metastasis processes.

**Table 1 ijms-21-03457-t001:** Surface markers of mouse and human preneutrophils, immature neutrophils and mature neutrophils. Lin, lineage markers.

	Mouse	Human
Pre-neutrophil	Lin^−^c-kit^int^CD11b^+^CXCR4^+^	Lin^−^CD66^+^CD15^+^CD33^med^ CD49d^med^CD101^−^
Immature neutrophil	Lin^−^c-kit^−^CD11b^+^Ly6G^+^ CXCR4^−^CXCR2^−^	Lin^−^CD66^+^CD15^+^CD33^med^ CD49d^−^CD101^med^CD10^−^CD16^med^
Mature neutrophil	Lin^−^c-kit^−^CD11b^+^Ly6G^+^ CXCR4^−^CXCR2^+^	Lin^−^CD66^+^CD15^+^CD33^med^ CD49d^−^CD101^med^CD10^+^CD16^high^

+, positive; −, negative.

**Table 2 ijms-21-03457-t002:** Characteristic surface phenotypes of MDSCs.

	Mouse	Human
Total MDSC	Gr-1^+^CD11b^+^	Not clearly determined
PMN–MDSC	CD11b^+^Ly6C^low^Ly6G^+^	CD14^−^CD11b^+^CD15^+^CD66b^+^
monocytic MDSC	CD11b^+^Ly6C^high^Ly6G^−^	CD14^+^CD11b^+^CD15^−^HLA-DR^low^
early MDSC	Not clearly determined	Lin^−^HLA-DR^−^CD33^+^

+, positive; −, negative.

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
