# Peer review of "Two-Faced Roles of Tumor-Associated Neutrophils in Cancer Development and Progression"

_ijms, 2020, doi:10.3390/ijms21103457_

Round 1
Reviewer 1 Report
In this review, the authors present a comprehensive overview of the role of neutrophils in tumor development and tumor progression. They discuss the role of the NLR (neutrophil-lymphocyte ratio) in cancer patients and elucidate to a complex role of the tumor-associated neutrophils in tumor progression and immunity. The review is well structured and very well written.
While the goal of the review is to portray neutrophils as a prognostic factor in cancer, it would be helpful to add a short paragraph about the potential role of neutrophils in the response to therapy/immunotherapy. What is known about tumor-associated neutrophils as a predictive factor in cancer patients?
Author Response
In this review, the authors present a comprehensive overview of the role of neutrophils in tumor development and tumor progression. They discuss the role of the NLR (neutrophil-lymphocyte ratio) in cancer patients and elucidate to a complex role of the tumor-associated neutrophils in tumor progression and immunity. The review is well structured and very well written.
While the goal of the review is to portray neutrophils as a prognostic factor in cancer, it would be helpful to add a short paragraph about the potential role of neutrophils in the response to therapy/immunotherapy. What is known about tumor-associated neutrophils as a predictive factor in cancer patients?
We appreciate thoughtful comments on our manuscript.
In response to the recommendation to add a short paragraph about the potential role of neutrophils in the response to therapy/immunotherapy, we added the following paragraph.
“TANs can be a cellular target of anti-cancer treatment measures: neutrophil depletion, inhibition of neutrophil accumulation in tumor sites, and modulation of neutrophil functional phenotypes. However, these measures have not been examined in clinical trials. For instance, the administration of anti-neutrophil antibody could deplete TANs in preclinical models [156] but it could simultaneously reduce circulating neutrophils, which have profound roles in diverse host responses to various insults, particularly bacterial infections [165]. Thus, neutrophil depletion can enhance susceptibility to bacterial infection and therefore, encounter difficulties in conducting a clinical trial.” (Lines 434 to 440)
In response to the question what is known about tumor-associated neutrophils as a predictive factor in cancer patients, we added the following sentences with several citations.
“Accumulating evidence indicates the association of a high density of TANs with poor prognosis of cancer patients [11], but this association did not apply to several types of cancers, such as squamous cell lung cancer [12] and colorectal cancer [13].” (Lines 51 to 53)
Major revisions are indicated in red in the revised text.
Reviewer 2 Report
The manuscript ijms-798152 presents an overview on tumor-associated neutrophils in cancer development and progression. The subject of the paper is not highly original, considering other similar reviews on this subject.
See for example:
Tumor-Associated Neutrophils in Cancer: Going Pro, Cancers (Basel). 2019 Apr; 11(4): 564
Tumor Associated Neutrophils. Their Role in Tumorigenesis, Metastasis, Prognosis and Therapy, Front. Oncol., 15 November 2019
The authors should highlight in introduction what their review has to offer new to other similar works. It would be also very good if the authors have a more critical presentation of the data and not just a collection of them.
The manuscript is well organized and presents a good amount of data on the subject. In some cases, newer references could be added, or changed the old ones with more recent data.
The article has multiple editorial problems. The authors should check the IJMS template and arrange the manuscript on it. For example, the manuscript is written in page 1 as “Int. J. Mol. Sci. 2018”. The abbreviations should be also checked and properly used.
The title is interesting, but it can be confusing. It sounds smart, but I’m not sure that many people knows the Roman mythology, and could confuse with JAK.
Author Response
The manuscript ijms-798152 presents an overview on tumor-associated neutrophils in cancer development and progression. The subject of the paper is not highly original, considering other similar reviews on this subject.
The authors should highlight in introduction what their review has to offer new to other similar works. It would be also very good if the authors have a more critical presentation of the data and not just a collection of them.
We appreciate very much the thoughtful comments.
In response to this recommendation, we modified several parts of manuscript, in order to make clear the aim of the present review article.
Abstract
“Here, we will discuss TAN biology in various tumorigenesis processes, particularly by focusing the context-dependent functional heterogeneity of TANs.” (Lines 23 to 24)
“Accumulating evidence indicates the association of a high density of TANs with poor prognosis of cancer patients [11], but this association did not apply to several types of cancers, such as squamous cell lung cancer [12] and colorectal cancer [13]. These context-dependent distinct roles of TANs may mirror their diverse actions under various tumor microenvironments. In this review article, after briefly summarizing the biology of neutrophils, we will discuss the functional heterogeneity of TANs in various tumorigenesis processes and the possible anti-cancer strategies targeting the diversity of TANs.” (Lines 51 to 57)
“Hepatocyte growth factor (HGF) is produced mainly by fibroblasts in tumor microenvironment, and mediates various tumorigenic processes including cell proliferation, survival, invasion, and angiogenesis, by acting on its specific receptor, Met [85]. In human brochioloalveolar subtype lung adenocarcinoma tissues, HGF and Met were detected in TANs and in cancer cells, respectively [86], suggesting that tumor progression can be provoked by TAN-derived HGF. This assumption may be supported by the observations that HCC cell-derived GM-CSF activated TANs to produce HGF and that the treatment with anti-neutrophil or anti-HGF antibodies reduced tumor formation [87]. On the contrary, TNF-a induced Met expression in neutrophils, which could release nitric oxide upon HGF stimulation, thereby killing tumor cells [88]. These observations suggest that TANs can exhibit context-dependent distinct roles in carcinogenesis through the actions of the HGF-Met axis.
TANs can have similar janus-faced roles in colon cancer model induced by the combined treatment of azoxymethane (AOM) and dextran sulfate sodium (DSS). In this model, a massive neutrophil infiltration was observed in colon cancer tissues [89] and anti-neutrophil antibody treatment decreased TANs which promoted carcinogenesis by providing MMP-9 [90]. On the contrary, genetically neutropenic mice were more susceptible to AOM/DSS-induced colon carcinogenesis in association with increased bacteria numbers in tumors [91], indicating that neutrophils can restrict bacteria growth and subsequently delay tumor development. This counteracting action of neutrophils may account for improved survival of stage II colorectal cancer patients with high levels of TANs [13].
Similar two-faced roles were observed on neutrophil extracellular traps (NETs) which are release by neutrophils and are composed of DNA coated with histone, NE, MPO, MMP-9, and cathepsin G [92]. NETs can inhibit melanoma cell migration and their proliferation, probably with the use of MPO [93]. On the contrary, tumor cells can induce the release of NETs [94], which can further promote tumor progression by inducing blood clot formation [95]. Moreover, NETs can further provide NE and/or MMP-9 to promote tumor cell growth [96]. NETs can further trap circulating tumor cells (CTCs) in vitro and in vivo [97,98], thereby facilitating their migration into distant organs.” (Lines 225 to 250)
“Diverse immune activities of TANs can be ascribed to the presence of functional dichotomy of TANs between N1- and N2-like ones, which can exhibit anti-tumorigenic and pro-tumorigenic functions, respectively [32]. N2-like phenotype can be maintained under the influence of tumor microenvironment, particularly hypoxia [36], which is more prevalent in advanced tumor sites than early ones [130]. Thus, TANs may exhibit distinct immune functions triggered by tumor environmental cues.” (Lines 321 to 326)
The manuscript is well organized and presents a good amount of data on the subject. In some cases, newer references could be added, or changed the old ones with more recent data.
In response to the recommendation, we changed the old reference 4 with a more recent review article (new references 2, 5, 14, 34). However, we did not replace original papers even though they were published long ago (e.g. references 8 and 173), due to our respect for the originality of these papers.
The article has multiple editorial problems. The authors should check the IJMS template and arrange the manuscript on it. For example, the manuscript is written in page 1 as “Int. J. Mol. Sci. 2018”. The abbreviations should be also checked and properly used.
We edited the manuscript in accordance with the template and checked the abbreviations again.
The title is interesting, but it can be confusing. It sounds smart, but I’m not sure that many people knows the Roman mythology, and could confuse with JAK.
We changed the title as follows, in order to avoid confusion.
“Two-faced roles of tumor-associated neutrophils in cancer development and progression”
Major revisions are indicated in red in the revised text.
Reviewer 3 Report
The authors have submitted a comprehensive review on neutrophil functions in cancer development and function. It is well written and includes illustrations that support the text.
My only suggestion will be to include caveats or pitfalls in studies that are discussed. Also it is important to include some perspective on what authors think will be discovered/researched in the future in this field. Understanding of some aspects of neutrophil biology may be more important than others and this should be illuminated towards the end of this manuscript.
Author Response
The authors have submitted a comprehensive review on neutrophil functions in cancer development and function. It is well written and includes illustrations that support the text.
My only suggestion will be to include caveats or pitfalls in studies that are discussed. Also it is important to include some perspective on what authors think will be discovered/researched in the future in this field. Understanding of some aspects of neutrophil biology may be more important than others and this should be illuminated towards the end of this manuscript.
We appreciate very much thoughtful comments on our manuscript.
In response to the recommendation, we added the following sentences.
“Nevertheless, it is still unclear on most biological aspects of TANs in human tumor tissues, particularly their specific functions in each type of human tumor tissue. As the conundrum will be solved with the use of recently developed analytical methods, such as single-cell RNAseq and/or mass cytometry, anti-cancer therapy targeting TANs will emerge as a promising strategy.” (Lines 476 t0 480)
Major revisions are indicated in red in the revised text.